# Local Inverse Mapping Implicit Hole-Cutting Method for Structured Cartesian Overset Grid Assembly

**DOI:** 10.3390/e25030432

**Published:** 2023-02-28

**Authors:** Jingyuan Wang, Feng Wu, Quanyong Xu, Lei Tan

**Affiliations:** 1State Key Laboratory of HydroScience and Engineering, Department of Energy and Power Engineering, Tsinghua University, Beijing 100084, China; 2AECC Sichuan Gas Turbine Establishment, Mianyang 621000, China; 3Institute for Aero Engine, Tsinghua University, Beijing 100084, China

**Keywords:** navier-stokes partial differential equations, local inverse mapping, coordinate transformation, X-ray method, implicit hole-cutting, overset grid assembly, structured cartesian mesh

## Abstract

An automatic hole-cutting method is proposed to search donor cells between a structured Cartesian mesh and an overlapping body-fitted mesh. The main flow is simulated on the structured Cartesian mesh and the viscous flow near the solid boundary is simulated on the body-fitted mesh. Through the spatial interpolation of flux, the surface boundary information on the body-fitted mesh is transferred to the Cartesian mesh nodes near the surface. Cartesian mesh box near a body-fitted mesh cell is selected as a local inverse map. The Cartesian nodes located inside the donor cells are marked by the relative coordinate transformation, so that all Cartesian nodes can be classified and the hole boundaries are implicitly cut. This hole-cutting process for overset grid assembly is called Local Inverse Mapping (LIM) method. In the LIM method, spatial interpolation of flux is carried out synchronously with the marking of Cartesian nodes. The LIM method is combined with the in-house finite-difference solver to simulate the unsteady flow field of moving bodies. The numerical results show that the LIM method can accurately mark the Cartesian hole boundary nodes, the search efficiency of donor cells is high, and the result of spatial interpolation is accurate. The calculation time of overset grid assembly (OGA) can be less than 3% of the total simulation time.

## 1. Introduction

Cartesian mesh is simple to generate, requires fewer computing resources, and requires no update for moving bodies. Peskin [1] proposed the immersed boundary method (IBM), which makes it feasible to calculate the flow field of objects with complex shapes in a Cartesian grid. The “Euler–Cartesian grid” is used to represent the fluid region and the “Lagrangian body surface grid” is used to represent the solid region. The effect of solids on fluids is realized through the construction and dispersion of “surface force sources”. By using the “spatial discrete function”, the Lagrangian force source term on the body surface grid nodes is extrapolated to a group of nearby Euler fluid Cartesian grid nodes, and the discrete Euler force source term is obtained. Then the Euclidean source term is put into the governing equations to solve the flow field to satisfy the boundary condition of the body surface.

Immersed boundary method has a simple force source structure and is easy to solve, but the mesh resolution near the wall surface is low, resulting in low solution accuracy of the flow field near the wall [2,3,4]. For viscous flow simulation with a high Reynolds number, a large number of grid nodes is needed to ensure adequate grid resolution near the wall. A hierarchical Cartesian grid is often used to improve the grid resolution and reduce grid node requirements. Chen Congcong et al. [5] states that a hierarchical Cartesian grid (as shown in Figure 1) can save 70% of computational grids compared to the simple stretch grid.

The overset grid method [6], which solves the viscous flow on the body-fitted mesh around the wall and solves the main flow on the Cartesian mesh, can retain the advantages of simple generation, low memory requirements and high computational efficiency of Cartesian mesh. The overset grid method consists of two core technologies: Overset Grid Assembly (OGA) and Interpolated Data Transfer (IDT). The OGA process marks the Cartesian grid nodes that overlap the body-fitted mesh, and finds out the donor cells for information transfer. The IDT between two sets of grids is realized by spatial interpolation of physical flux.

The traversal method [7,8] is the most common OGA method used to identify the positional relationship between the body-fitted mesh and the Cartesian mesh. In the case of three dimensions of the traversal method, a very small Cartesian mesh spacing will result in a sharp increase in the computation consumption. Stencil Walk method or Stencil Jump method [9], and Neighbor-to-Neighbor Search method [10,11] are improved traversal methods, which can increase the computational efficiency of OGA to some extent but still require a huge amount of computation for 3D meshes. The Wall-Distance method [12,13] reduces the number of donor cells to be searched, thus reducing the amount of computation. When the number of Grid nodes exceeds 10 million, parallel computing method [14,15,16] should be adopted for Overset Grid Assembly to improve computing efficiency. The alternating digital tree (ADT) algorithm [17,18] applied to the adaptive Cartesian grid can effectively reduce the computation cost of OGA, because of its efficient data query structure. However, such methods cannot be applied to the structured Cartesian mesh.

In the structured overlapping Cartesian grid, WENO [19] scheme and other high-order finite difference schemes can be used to give full play to the advantages of the Cartesian grid. NASA carried out a series of studies in the 1980s and 1990s, and proposed hole mapping method [20], inverse map method [21] and X-Ray method [22], which significantly improved the OGA computing efficiency of the structured overset grid. At present, the research progress of the structured overset grid assembly method is not much, except Exact Inverse Map (EIM) method [23].

In order to improve the computational efficiency of structured overset grid assembly, we proposed an implicit hole cutting (IHC) method [24,25] named Local Inverse Mapping (LIM) method, for fluid points near the surface (FPNS, as shown in Figure 2) marking and flux interpolation between Cartesian mesh and overlapping body-fitted mesh. Structured Cartesian mesh is used as a reference scale, Newton iteration method [26] and local inverse map are used to calculate the relative position between a given point and a donor cell of body-fitted mesh. Thus, marking the Cartesian mesh nodes near the body surface. At the same time, spatial interpolation [4] is carried out on FPNS points.

## 2. Implicit Hole-Cutting Procedure by LIM Method

Figure 2 shows a Cartesian mesh block consisting of nine points (yellow dotted line range) surrounding a hexahedral cell of the body-fitted cylinder mesh. This Cartesian grid block is defined as a “local inverse mapping box”. The relative coordinates of those nine points of the local inverse mapping box are calculated, respectively. If the relative coordinate value of a point is within the range of [0, 1], this point is marked as Fluid Point Near Surface (FPNS). The flux at the FPNS point needs to be interpolated from the donor cell of the body-fitted mesh by Lagrange polynomial interpolation formula. Repeat the above steps to traverse all hexahedral donor cells of body-fitted mesh, thus all FPNS points on the Cartesian mesh are marked. Finally, through an X-Ray method, all cartesian mesh nodes located inside the solid surface are marked.

### 2.1. Strategies for Boundary Information Exchange

On one time-step “*t*”, a suitable “donor cell” of the Cartesian grid (the red quadrilateral in Figure 2), surrounding an extra boundary point of the body-fitted grid is found, and the flow field information on that “donor cell” of the Cartesian grid is interpolated into the boundary point of the body-fitted grid as the “additional boundary condition”;By solving the boundary layer flow field near the surface on the body-fitted grid, and then the information of the surface force at time-step “t+dt/2” can be obtained;The fluid points on the Cartesian gird overlapping with the body-fitted grid are defined as fluid points near the surface (FPNS, as shown in Figure 2). Searching the “donor cell” of the body-fitted grid (the yellow quadrilateral in Figure 2) surrounding an FPNS point. Then, the surface flow information at the “t+dt/2” time-step on the body-fitted grid is interpolated to the FPNS point through the spatial interpolation formula. Repeat the above step to obtain surface flow information at all FPNS points;Finally, the updated surface flow information on FPNS points is used as an “additional boundary condition” to solve the flow field on the whole Cartesian grid, and the solution of the time-step “t+dt” is completed.

### 2.2. Flux Interpolation Methods

Taking the donor cell hexahedron shown in Figure 3 as an example, the interpolated physical flux $\phi_{P}$ at point P can be calculated according to the interpolation formula [4]:(1)$$\phi_{P}=\underset{i=0}{\sum^{1}}\;\underset{j=0}{\sum^{1}}\;\underset{k=0}{\sum^{1}}\;\phi_{i,j,k}C_{i,j,k}\left(\xi ,\eta ,\zeta \right)=f_{\phi }\left(\phi_{\left(i,j,k\right)},\xi_{P},\eta_{P},\zeta_{P}\right)$$
where, $\phi_{\left(0∼1,0∼1,0∼1\right)}$ is the physical flux at a vertice of the donor cell, which is under the coordinate system (ξ,η,ζ).

According to $\left(\xi_{P},\eta_{P},\zeta_{P}\right)$, the coordinates of point *P*, the interpolation coefficients $C_{i,j,k}\left(\xi ,\eta ,\zeta \right)$ at eight vertices can be calculated as follows:(2)$$\left\{\begin{array}{c} \left(1-\xi \right)\left(1-\eta \right)\left(1-\zeta \right)\\ \xi \left(1-\eta \right)\left(1-\zeta \right)\\ \xi \eta \left(1-\zeta \right)\\ \left(1-\xi \right)\eta \left(1-\zeta \right)\\ \left(1-\xi \right)\left(1-\eta \right)\zeta \\ \xi \left(1-\eta \right)\zeta \\ \xi \eta \zeta \\ \left(1-\xi \right)\eta \zeta \end{array}\right.\begin{array}{c} i,j,k=\left(0,0,0\right)\\ i,j,k=\left(1,0,0\right)\\ i,j,k=\left(1,1,0\right)\\ i,j,k=\left(0,1,0\right)\\ i,j,k=\left(0,0,1\right)\\ i,j,k=\left(1,0,1\right)\\ i,j,k=\left(1,1,1\right)\\ i,j,k=\left(0,1,1\right) \end{array}$$

### 2.3. Strategies for Donor Cells Searching and Cartesian Nodes Classification

For a Cartesian mesh point P in the local inverse mapping box as shown in Figure 2, its spatial position relationship between itself and the donor cell can be determined by solving the nonlinear equations and obtaining the relative coordinates. If the relative coordinate value of point P, $\left(\xi_{P},\eta_{P},\zeta_{P}\right)$ is within the range of [0, 1], it indicates that this point is inside the donor cell.

The nonlinear equations are established to solve the relative coordinates $\left(\xi_{P},\eta_{P},\zeta_{P}\right)$ of point P, by given the Cartesian coordinates $\left(x_{P},y_{P},z_{P}\right)$ of point P, and the Cartesian coordinates $x_{\left(i,j,k\right)}$, $y_{\left(i,j,k\right)}$, $z_{\left(i,j,k\right)}$, $i,j,k\in \left\{0,1\right\}$ of 8 vertex points of the donor cell, and substituting these coordinates of nine points into Equation (Equation 1). The form of this nonlinear system is as follows:(3)$$\left\{\begin{array}{c} x_{P}=f_{x}\left(x_{\left(i,j,k\right)},\xi_{P},\eta_{P},\zeta_{P}\right)\\ y_{P}=f_{y}\left(y_{\left(i,j,k\right)},\xi_{P},\eta_{P},\zeta_{P}\right)\\ z_{P}=f_{z}\left(z_{\left(i,j,k\right)},\xi_{P},\eta_{P},\zeta_{P}\right) \end{array}\right.$$

### 2.4. X-ray Method and Cartesian Hole-Cutting Results

The FPNS Cartesian nodes marked by the LIM method are shown as the lower left vertex of the red square in Figure 4a. The Cartesian solid nodes inside the cylinder are marked by the X-ray method. Take the leftmost boundary of the Cartesian mesh as a starting point and move the unmarked point to the right along the ray. If the adjacent point to the left of the moving point is marked and the point to the right is unmarked, then this point is recorded as the starting point. Continue to move the unmarked point to the right along the ray until the adjacent point to the left of the moving point is unmarked and the point to the right is marked, then record the point as the end point. Mark all points between the start point and the end point as solid points. Both the start and end points are marked as ghost-cells. The last end point is then used as the new ray starting point, and the preceding steps are repeated until all the inner solid points are marked.

The solid points marked by the X-Ray method are shown in the lower left vertex of the blue square in Figure 4b, while the ghost cell points are shown in the lower left vertex of the green square in Figure 4. The donor cells searching method and flux interpolation method adopted by LIM are simple and efficient. The computation cost of spatial interpolation and information exchange could be less than 3% of the flow field simulation.

### 2.5. Information Entropy Analysis of OGA Computing Process

Information entropy can be used to characterize the complexity of OGA computation. The more complex the OGA process is, the more redundant computation is involved and the higher the information entropy is. Information theory provides a theoretical foundation to quantify the information content, or the uncertainty, of a random variable represented as a distribution. Formally, let X be a discrete random variable with alphabet χ and probability mass function p(x), x ∈χ. The Shannon entropy of X is defined as [27]
(4)$$H\left(X\right)=-\sum_{x\in \chi }\;p\left(x\right)\log\left[p\left(x\right)\right]$$

The key to applying the concept of entropy to OGA problems lies in how to specify a suitable random variable *X* and define the probability function p(x). In most cases, these probability functions can be defined heuristically to suit the needs of individual applications [28]. The probability p(x) of the OGA calculation process can be defined as the probability that any cell on a body-fitted grid is a donor cell of a background grid node. Assuming that the scale of the background grid is the same as that of the body-fitted grid, the total number of nodes in the background grid is $n_{b}$, and the total number of cells in the body-fitted grid is $n_{a}$, then the probability mass function p(x) corresponding to the traversal method is as follows
(5)$$p\left(x\right)=\frac{1}{n_{a}n_{b}}$$
Shannon entropy of the traversal method is
(6)$$H\left(X_{1}\right)=-\underset{i=1}{\sum^{n_{a}n_{b}}}\;\frac{1}{n_{a}n_{b}}\log\left(\frac{1}{n_{a}n_{b}}\right)=\log\left(n_{a}n_{b}\right)$$
Under the same assumption, since a body-fitted grid cell only needs to be judged by the 27 Cartesian background grid nodes surrounding it, the probability mass function p(x) corresponding to the LIM method is as follows
(7)$$p\left(x\right)=\frac{1}{27n_{a}}$$
Shannon entropy of the LIM method is
(8)$$H\left(X_{2}\right)=-\underset{i=1}{\sum^{27n_{a}}}\;\frac{1}{27n_{a}}\log\left(\frac{1}{27n_{a}}\right)=\log\left(27n_{a}\right)$$
The information entropy difference between the traversal method and the LIM method is as follows
(9)$$\Delta H=H\left(X_{1}\right)-H\left(X_{2}\right)=-\log\left(27/n_{b}\right)$$
When $n_{b}$ is much larger than 27, the information entropy of the traversal method will be much larger than that of the LIM method. Therefore, through information analysis, it can be seen that the LIM method has lower information entropy, less redundant calculation and higher computational efficiency than the traversal method. Especially for the 3D case with a large number of background grid nodes, the difference in OGA computational efficiency is more obvious.

## 3. Numerical Solver

The LIM method is added to the in-house code developed by the authors for computational fluid dynamics simulation. The governing equations, spatial and temporal discrete methods, boundary conditions and turbulence models used in the CFD program are briefly introduced in this chapter.

### 3.1. Governing Equations

Governing equations on the fixed orthogonal Cartesian grid and the movable body-fitted grid is the compressible Navier–Stokes equations, which can be expressed in the Cartesian coordinate system (x,y,z,t) as follows [29]:(10)$$\frac{\partial U}{\partial t}+\frac{\partial F}{\partial x}+\frac{\partial G}{\partial y}+\frac{\partial H}{\partial z}=\frac{\partial F_{v}}{\partial x}+\frac{\partial G_{v}}{\partial y}+\frac{\partial H_{v}}{\partial z}$$
where, the vector U is the conserved flux, the vector F,G,H are the inviscid flux terms (convection terms), and vector $F_{v},G_{v},H_{v}$ are the viscous flux terms (dissipative terms).

The expressions of the above 7 vectors are as follows:$$F=\left[\begin{array}{c} \begin{array}{c} \rho u_{r}\\ \rho uu_{r}+p\\ \rho vu_{r}\\ \rho wu_{r}\\ \left(\rho e+p\right)u_{r}+pu_{e} \end{array} \end{array}\right],F_{v}=\left[\begin{array}{c} \begin{array}{c} 0\\ \tau_{xx}\\ \tau_{xy}\\ \tau_{xz}\\ u\tau_{xx}+v\tau_{xy}+w\tau_{xz}+\kappa \frac{\partial T}{\partial x} \end{array} \end{array}\right]$$
$$G=\left[\begin{array}{c} \begin{array}{c} \rho v_{r}\\ \rho uv_{r}\\ \rho vv_{r}+p\\ \rho wv_{r}\\ \left(\rho e+p\right)v_{r}+pv_{e} \end{array} \end{array}\right],G_{v}=\left[\begin{array}{c} \begin{array}{c} 0\\ \tau_{yx}\\ \tau_{yy}\\ \tau_{yz}\\ u\tau_{yx}+v\tau_{yy}+w\tau_{yz}+\kappa \frac{\partial T}{\partial y} \end{array} \end{array}\right]$$
$$H=\left[\begin{array}{c} \begin{array}{c} \rho w_{r}\\ \rho uw_{r}\\ \rho vw_{r}\\ \rho ww_{r}+p\\ \left(\rho e+p\right)w_{r}+pw_{e} \end{array} \end{array}\right],H_{v}=\left[\begin{array}{c} \begin{array}{c} 0\\ \tau_{zx}\\ \tau_{zy}\\ \tau_{zz}\\ u\tau_{zx}+v\tau_{zy}+w\tau_{zz}+\kappa \frac{\partial T}{\partial z} \end{array} \end{array}\right]$$
(11)$$U={\left(\rho ,\rho u,\rho v,\rho w,\rho e\right)}^{T}$$
where, ρ is the density, *p* is the pressure, *T* is the temperature, *e* is the total energy, τ is the viscosity stress and κ is the thermal conductivity. ${\left(u,v,w\right)}^{T}$ is the velocity vector under the coordinate system $\left(x,y,z\right)$, ${\left(u_{r},v_{r},w_{r}\right)}^{T}$ is the relative velocity vector, and ${\left(u_{e},v_{e},w_{e}\right)}^{T}$ is the convected velocity vector of the movable body-fitted grid.

### 3.2. Spatial Discretization

Because of the strong hyperbolic property of convection terms in N-S equations, upwind schemes, such as the WENO scheme, which has the characteristics of high discontinuity resolution, high difference accuracy, slight numerical oscillation and small numerical dissipation, should be selected. Therefore, our in-house CFD code adopts flux-vector splitting schemes [6] combined with 5th-order WENO scheme [19,30] to carry out the spatial dispersion of convection terms. The 6th-order central difference scheme is used for spatial discretization of dissipative terms due to its dissipative property.

### 3.3. Temporal Discretization

The explicit three-step and third-order R-K method [31] is adopted. Since two sets of nonuniform meshes, background meshes and surface meshes, are, respectively, used to solve the mainstream and boundary layer flows, the two sets of meshes can adopt different global time steps. A small local time step is used on the surface body-fitted grid and a large time step is used on the Cartesian grid. Every step of the flow field solution is pushed forwards in time on the Cartesian grid, and several steps on the body-fitted grid, so as to improve the computational efficiency. This time-step strategy works in a similar way as Jameson’s dual time-step approach [32].

### 3.4. Boundary Conditions and Turbulence Models

Whitfield and Janus’s “non-reflection boundary condition” [33] derived from Euler equations based on the characteristic theory, is applied to all boundaries of the Cartesian grid. While “Non-slip wall boundary condition” is adopted on the solid surface. Spalart-Allmaras turbulence model [34] is used for viscous flow simulation.

## 4. Numerical Results

In order to verify the LIM method, an in-house CFD solver was used to simulate the flow around a 2-D cylinder and contra-rotating open rotors.

### 4.1. Fixed 2-D Cylinder

#### 4.1.1. Mesh Domain and Simulation Conditions

The 2-D structured computational domain is shown in Figure 5. This domain has been discretized with the nonuniform Cartesian grid. The grid spacing Δx and Δy is uniform in the region of the Cartesian grid containing the cylinder (as shown by the close-up view in Figure 5). Note that the finest mesh domain will cover the area where the oscillating cylinder moves if the cylinder is movable. The grid spacing is smoothly stretched from those values to near the inflow, outflow, and top and bottom boundaries. The computational domain is [−3D: 20D] × [−5D: 5D], where *D* is the diameter of the cylinder. The center of the cylinder is located at the point (0.5D, 0D) of the coordinate system. Due to the application of non-reflection boundary conditions, there is no need to set the computational domain rather large to reduce wave reflections from the left and right boundaries.

The boundary conditions of this computational domain are:Nonreflection boundary condition are applied at the top and bottom;Nonslip wall boundary condition is implemented on the cylinder surface;Nondimensional speed $U_{\infty }=1$, Mach number Ma = 0.1, and pressure *p* = 101,325 Pa, temperature *T* = 300 K for far-field flow;Velocity is a uniform flow at the left boundary and zero-gradient at the right boundary.

The time-evolution of the flow field can be quantitatively represented by the variation in the drag and lift coefficients at the surface of the body, which are defined as:(12)$$C_{d}=\frac{F_{d}}{\frac{1}{2}\rho_{\infty }U_{\infty }^{2}D},\;C_{L}=\frac{F_{L}}{\frac{1}{2}\rho_{\infty }U_{\infty }^{2}D}$$
where *D* is the diameter of cylinder, $\rho_{\infty }$ is the free-stream density, and $U_{\infty }$ is the free-stream velocity, $F_{d}$ is the drag force and $F_{L}$ is the lift force acting on the circular cylinder.

The dynamic force F acting on the surface of the cylinder can be computed by integrating the local pressure and stress distributions along the cylinder surface, which can be written as:(13)$$F=\left(F_{x},F_{y}\right)=\left(F_{d},F_{L}\right)=\oint_{S_{b}}\left(-p_{b}I+\tau_{b}\right)\cdot nds$$
where $p_{b}$ is the pressure, I is the identity matrix, $n=(n_{x},n_{y})$ is the surface unit normal vector, and the subscripts *x* and *y* denote the x-direction and y-direction, respectively. τ is the stress tensor and $S_{b}$ is the surface of the solid body.

#### 4.1.2. Steady Flow Past a Fixed Cylinder at Re = 40

This case is steady laminar flow past a circular cylinder at Re = 40 and the freestream Mach number Ma = 0.1. Figure 6 is the result of velocity magnitude and streamlines of flow past the fixed cylinder. $L_{w}/D$ is the non-dimensional wake bubble length. Table 1 shows the good agreement in the drag coefficient $\bar{C_{d}}$ and $L_{w}/D$.

#### 4.1.3. Unsteady Flow Past a Fixed Cylinder at Re = 185

To investigate the sensitivity of the results to grid resolution, unsteady flow past a stationary circular cylinder at Re = 185 and Ma = 0.1 is simulated with three different grids spacings, which is Δx = Δy = 0.01D, 0.02D, 0.04D, 0.08D, respectively. Coefficients of $\bar{C_{d}}$ and St with different $\Delta x_{min}$ are shown in Table 2. According to the comparison of the results in Table 2, the grid spacing Δx of the subsequent simulations in this paper is equal to 0.02D. The Strouhal number St is defined as follows:(14)$$S_{t}=\frac{f_{0}D}{U_{\infty }}$$
where $f_{0}$ is the vortex natural shedding frequency.

Figure 7 presents the drag and lift coefficients for this case. The average drag coefficient, ${\bar{C}}_{d}$, the root-mean-square value of the lift coefficient, $C_{Lr.m.s.}$, and the Strouhal number St are reported in Table 3. As shown in Table 3, St is in good agreement with Williamson’s universal law [38], and the flow parameters agree with other experimental and numerical results very well. From this, the value of $f_{o}$ can be calculated and is used in the following simulation for the transversely oscillating cylinder.

### 4.2. Oscillating 2-D Cylinder

The problem of a cylinder transversely oscillating in the free stream is a classic flow problem with moving boundaries, which can be used to verify the accuracy of the LIM method applied to OGA and flow field interpolation.

There are various experimental and numerical investigations available (Khalili et al. [2], Guilmineau and Queutey [39], Lu and Dalton [40], Wu [42], Gu et al. [43], Schneiders et al. [44]). The above studies showed that the vortex shedding frequency can synchronize with the frequency of the oscillatory forcing. This so-called lock-in of the vortex shedding is usually observed when the forcing frequency approaches the natural shedding frequency of the flow past a fixed cylinder.

The Reynolds number based on the cylinder diameter D and the freestream velocity u is Re = 185. The vertical cylinder position is imposed as $y\left(t\right)=A_{e}sin\left(2\pi f_{e}t\right)$ with $A_{e}$ and $f_{e}$ being the oscillation amplitude and the excitation frequency, respectively. The quantity $f_{0}$ denotes the natural vortex-shedding frequency obtained for the corresponding flow past a fixed cylinder. $f_{e}$ is very important for the flow structure. LIM method is used to perform calculations for Re=185, $A_{e}=0.2D$. The range of $f_{e}/f_{0}$ = 0.8, 0.9, 1.0, 1.1, 1.12, 1.2 is considered. All the simulations start from the flow past a non-oscillating cylinder until the vortex shedding reaches to the stable state.

#### 4.2.1. Drag and Lift Coefficients Simulation Results

Figure 8 shows the drag and lift coefficients for different values of $f_{e}/f_{0}$ obtained by the LIM method and WENO CFD solver. The primary changes in the drag and lift coefficient patterns can be seen as $f_{e}/f_{0}$ increases. The impact of the higher harmonic oscillations can be easily seen in this figure for excitation frequencies higher than $f_{e}/f_{0}$ ≥ 1. When $f_{e}/f_{0}$ ≤ 1.0, the force behaves periodically in time because stable vortex shedding is established. However, periodicity is lost for $f_{e}/f_{0}$ ≥ 1.1. Increasing $f_{e}$($f_{e}/f_{0}$ ≥ 0.9), the amplitudes for the lift coefficients are increasing. Since an opposite sign vortex is formed at the lower side of the cylinder as $f_{e}/f_{0}$ increases ($f_{e}/f_{0}$ > 1), as can be seen in Figure 9, this phenomenon causes changes in the vorticity at the top and bottom of the shear layers. The above results show good agreement with reference [39,43].

Table 4 and Table 5 compare the present time-averaged drag coefficient and the root-mean-square values of the drag and lift coefficients at $f_{e}/f_{0}$ = 0.8 and 1.1, respectively, with numerical simulation results from other references. It is found that they agree very well with each other.

#### 4.2.2. Flow Field Simulation Results

Figure 9 presents the instantaneous streamlines with velocity in the left column and instantaneous spanwise vorticity contours in the right column. Three cases are shown where $f_{e}/f_{0}$ is gradually increased from 0.8 to 1.12. It is observed that the wake pattern is suddenly changed from $f_{e}/f_{0}$ = 1.0 to $f_{e}/f_{0}$ = 1.12.

Figure 10 and Figure 11 are the comparison of the surface pressure coefficient for the cylinder located at the top-dead-center with the results of Guilmineau and Queutey [39]. Comparison results show good agreement of the distribution of the coefficient of pressure.

The above simulation results show that the LIM method can accurately implement the assembly of the moving overlapping mesh, and ensure the correct flow field interpolation and simulation results.

### 4.3. Contra-Rotating Open Rotors

In order to further verify the accuracy and reliability of LIM method, the overset grid assembly calculation and unsteady flow simulation of the Contra-rotating open rotors are carried out.

#### 4.3.1. OGA Calculation Efficiency Comparison

Figure 12 shows the surface mesh and body-fitted mesh of the Contra-rotating open rotor. Each blade block is an independent structured body-fitted grid with a node number of 0.204 million. The front and rear rotors have a total of 14 blades and 2.8 million grid nodes. Table 6 shows the number of nodes in the two sets of grids. The percentage of CPU time-consuming ratio of OGA is shown in Table 7. OGA computing is implemented on a personal computer with AMD Ryzen 3100 CPU. The traversal method and LIM method were used, respectively, to mark the FPNS Cartesian mesh points. The comparison of OGA calculation time is shown in Table 8. The comparison results show that the LIM method can reduce the computing time by 99% compared with the traversal method. Compared with the 2-D case, the computational efficiency of the LIM method is significantly improved in the 3-D case. Cartesian nodes overlapped with blade mesh were marked green by the LIM method as shown in Figure 13.

This example demonstrates the applicability of the LIM method to the multibody case. The front and rear rotors rotate in opposite directions. Each blade grid block can move independently, and there are 14 moving blocks in total. There is no exchange of flow information between the blade grid blocks, and the flow information can be transmitted only through flux interpolation with the Cartesian mesh.

#### 4.3.2. Flux Interpolation Strategy

High-Order Interpolation Method [47] and Conservative Interpolation Method [48,49,50] are very helpful to ensure the accuracy of simulation results of the overlapping grid. For structured grids, non-conservation interpolation [51] can also obtain satisfactory simulation results. The Lagrange polynomial spatial interpolation method used in this paper belongs to non-conservation interpolation. The physical quantity transferred by spatial interpolation is the vector U in Equation (Equation 11), rather than the original physical variables, such as density ρ and velocity u,v,w. Figure 14 shows the vorticity distribution unsteady simulation results of Contra-rotating open rotors. Figure 15 shows the simulation results of shock waves in an open rotor blade passage. The continuous vorticity and shock wave distribution verify that the LIM method is accurate and the spatial interpolation results are correct.

#### 4.3.3. Parallel Computing Strategy of LIM

LIM method can support both Coarse-grained and Fine-grained parallel computing strategies. In the Coarse-grained parallelism strategy, each moving grid block, such as the blade grid block, occupies one thread and performs OGA calculation. For 14 blade multi-body grid blocks, 14 threads can be used for parallel computation. In the Fine-grained parallelism strategy, the structured body-fitted blade mesh can be divided into np sub-blocks (np = number of threads) according to the number of threads. LIM method is used to mark the FPNS nodes on each sub-block in parallel. After the OGA parallel calculation on all sub-blocks is completed, the nodes in the Cartesian mesh that overlap with the body-fitted mesh are marked, and the flux is interpolated from the blade mesh donor cells. OGA parallel computing code is under development, and the efficiency improvements will be compared in subsequent studies.

## 5. Discussion and Conclusions

Compared with the traversal OGA method, the local inverse mapping implicit hole-cutting method proposed in this paper can automatically perform the overset grid assembly calculation without manual intervention. Verification results show that this method is efficient and accurate. The calculation cost of points marking and flux interpolation accounts for 3% of the total calculation cost for the 2-D case and 14% for the 3-D case. This method could effectively reduce the computing consumption of donor cells searching and save memory.

## Figures and Tables

**Figure 1 entropy-25-00432-f001:**
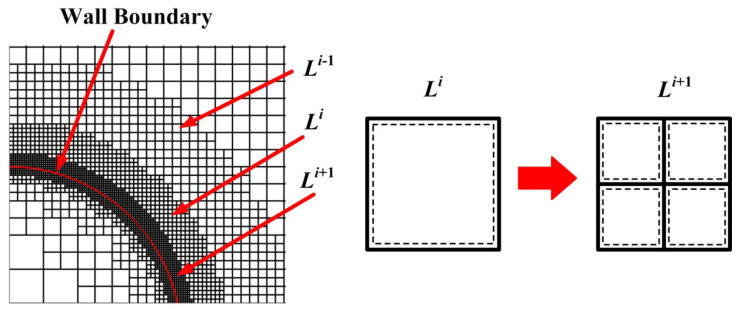
Hierarchical Cartesian grid of arbitrary wall. The solid line in red represents the wall boundary [5].

**Figure 2 entropy-25-00432-f002:**
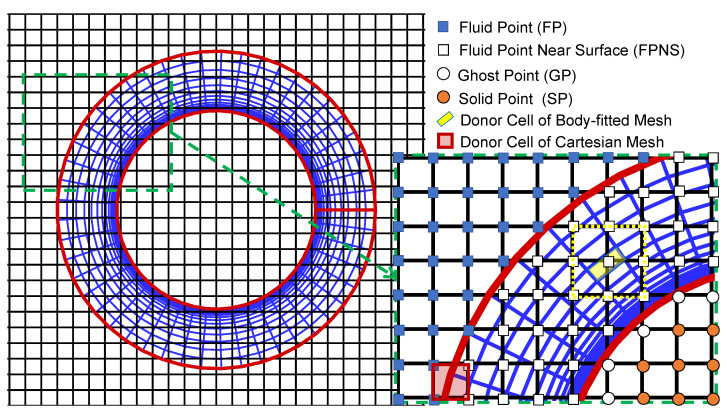
Nodes classification of body-fitted grid and Cartesian grid.

**Figure 3 entropy-25-00432-f003:**
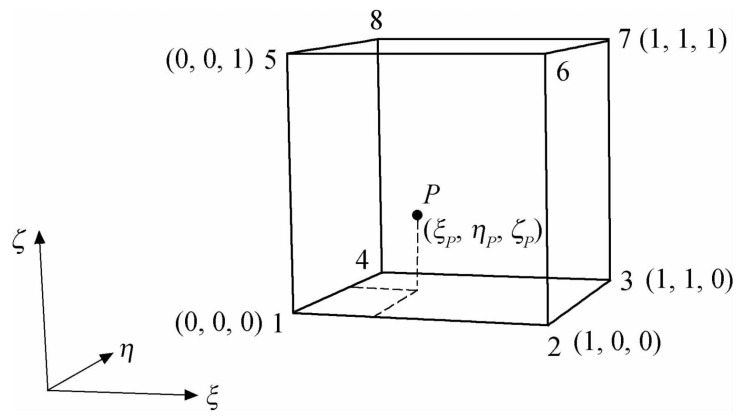
The interpolated point *P* and eight vertices in a donor cell, under the coordinate system (ξ,η,ζ).

**Figure 4 entropy-25-00432-f004:**
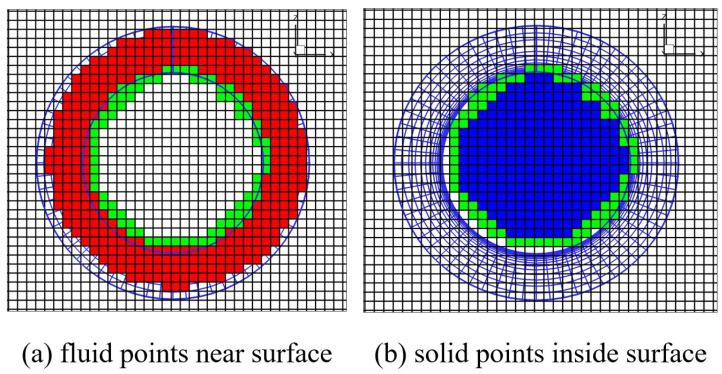
Nodes classification results of a 2-D circular cylinder grid.

**Figure 5 entropy-25-00432-f005:**
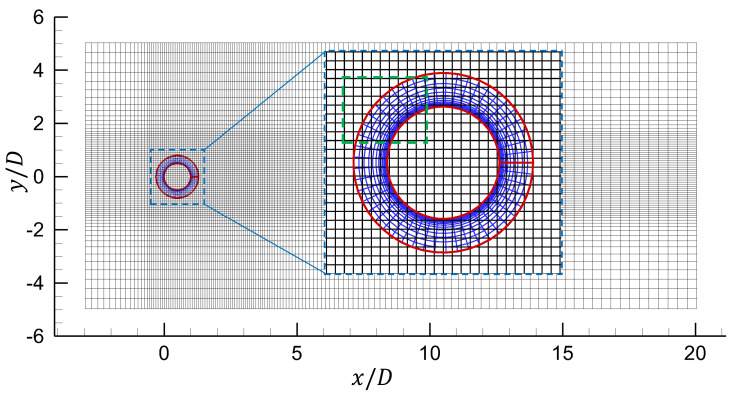
Computational Domain contain a circular cylinder ($\Delta x_{min}$ = 0.08D).

**Figure 6 entropy-25-00432-f006:**
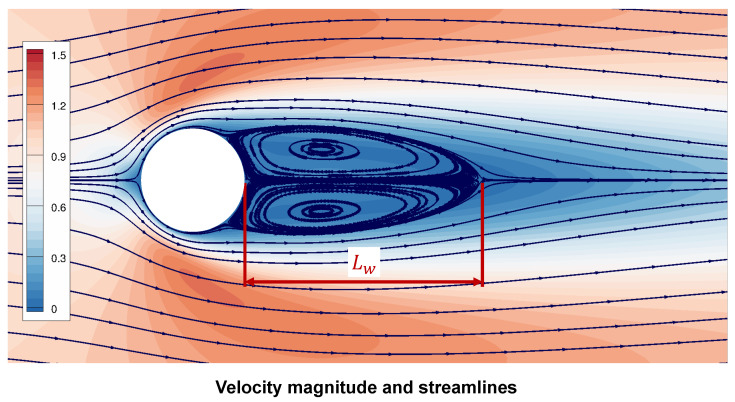
Streamlines of flow past a fixed cylinder at Re = 40.

**Figure 7 entropy-25-00432-f007:**
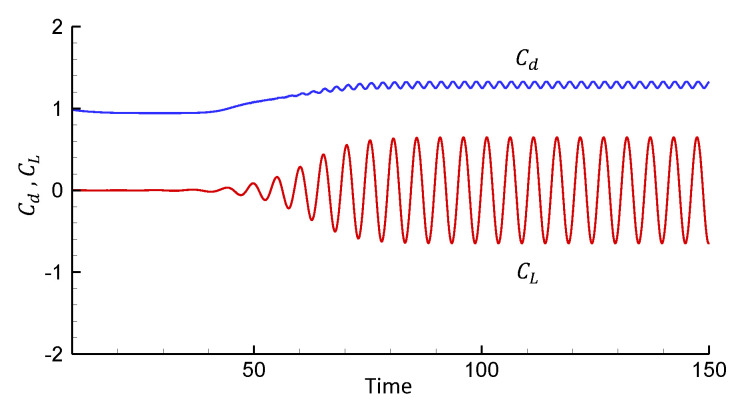
Lift and drag coefficients for a non-oscillating cylinder at Re = 185.

**Figure 8 entropy-25-00432-f008:**
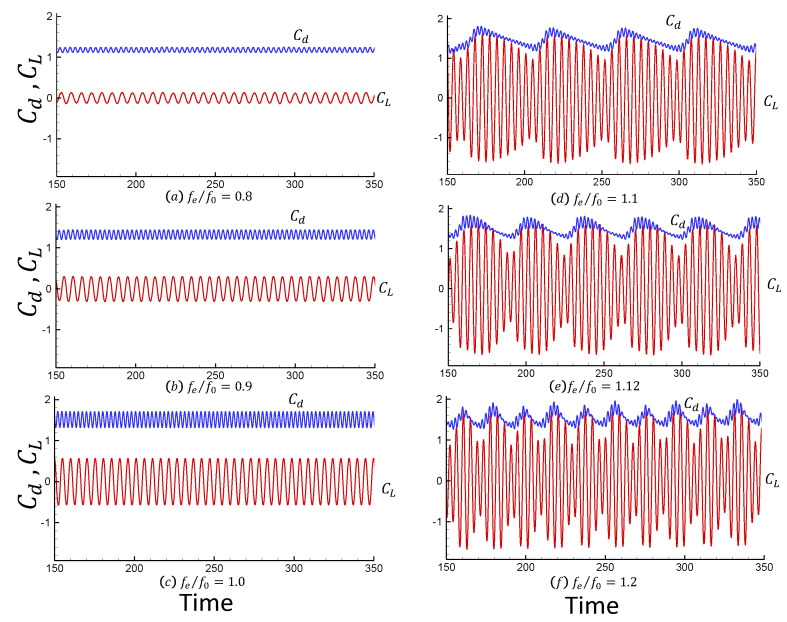
Drag and lift coefficients versus time for Re=185 and $A_{e}/D$ = 0.2 for values of $f_{e}/f_{o}$ equal to: (**a**) 0.80; (**b**) 0.90; (**c**) 1.00; (**d**) 1.10; (**e**) 1.12; (**f**) 1.20.

**Figure 9 entropy-25-00432-f009:**
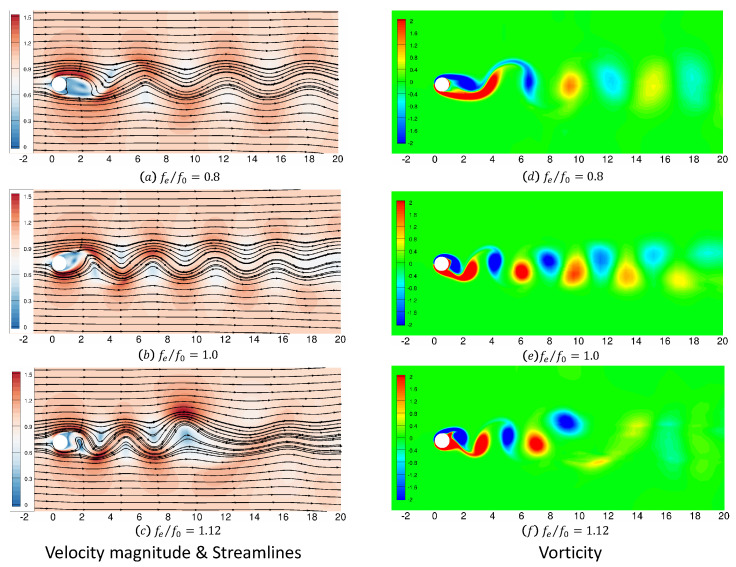
Velocity magnitude, streamlines and vorticity distributions of flow past a transversely oscillating cylinder at Re = 185 and Ma = 0.1 versus of $f_{e}/f_{0}$ = 0.8, 1.0 and 1.12.

**Figure 10 entropy-25-00432-f010:**
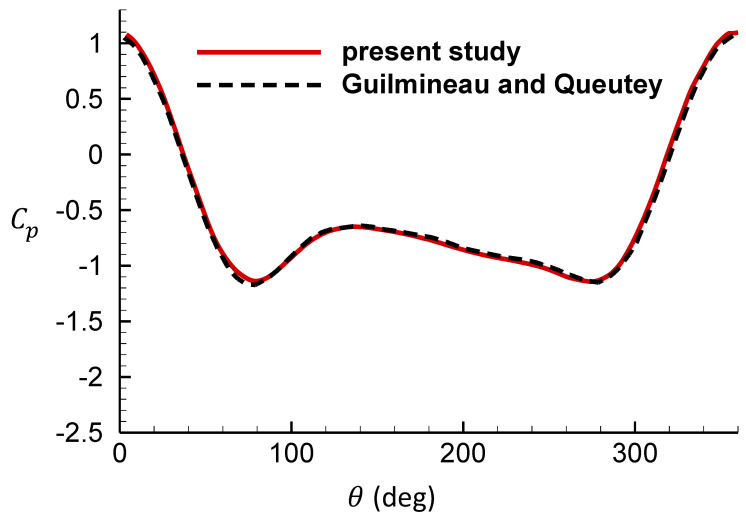
Comparison of the surface pressure coefficient for the cylinder located at the top-dead center against reference [39], $f_{e}$ = 0.8.

**Figure 11 entropy-25-00432-f011:**
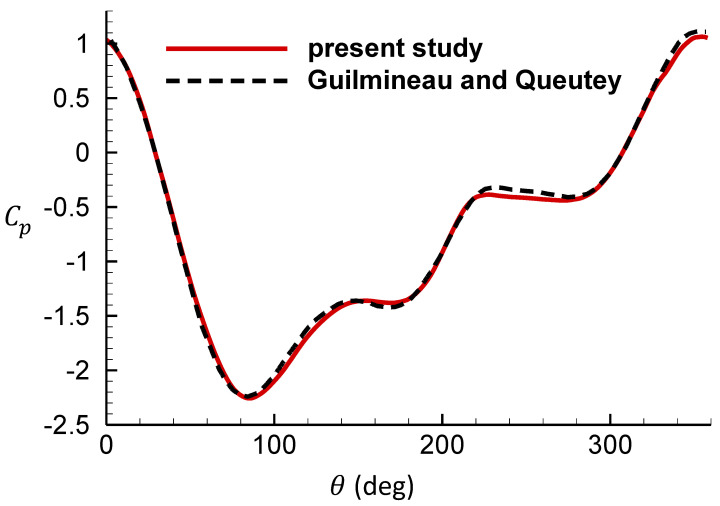
Comparison of the surface pressure coefficient for the cylinder located at the top-dead center against reference [39], $f_{e}$ = 1.1.

**Figure 12 entropy-25-00432-f012:**
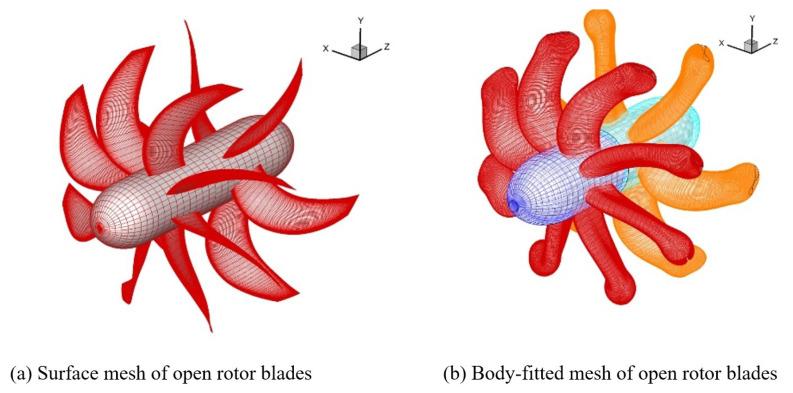
The surface mesh and body-fitted mesh of Contra-rotating open rotors.

**Figure 13 entropy-25-00432-f013:**
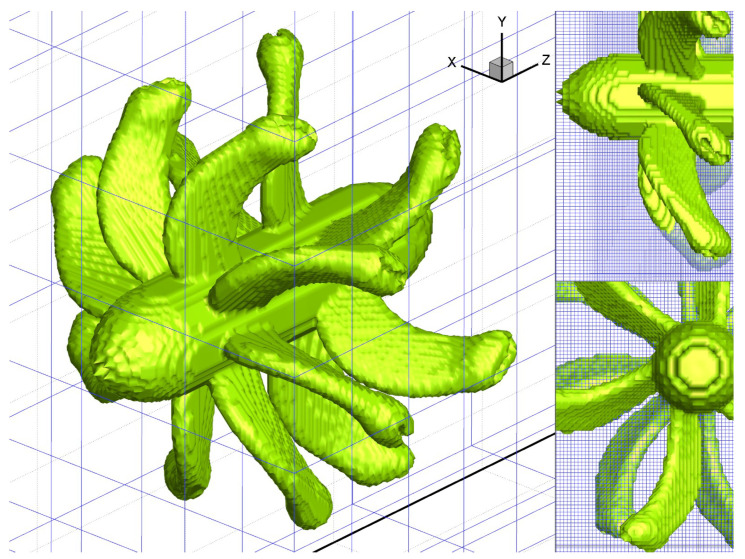
Marking results of Cartesian nodes overlapped with blades mesh by LIM method.

**Figure 14 entropy-25-00432-f014:**
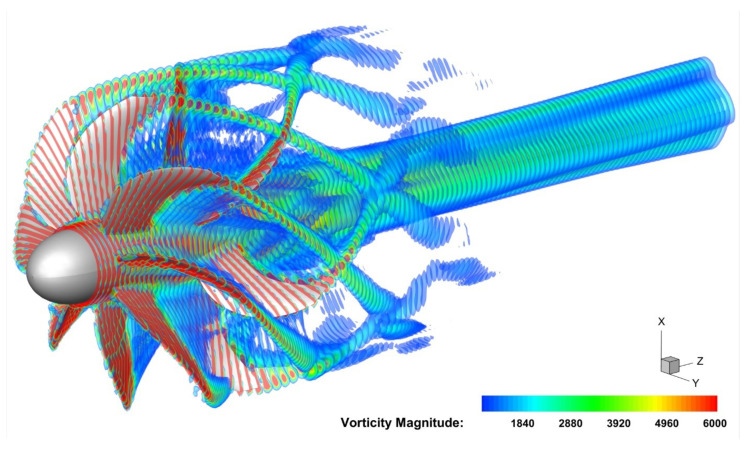
Vorticity results of Contra-rotating open rotors unsteady Simulation (three cycles of rotation).

**Figure 15 entropy-25-00432-f015:**
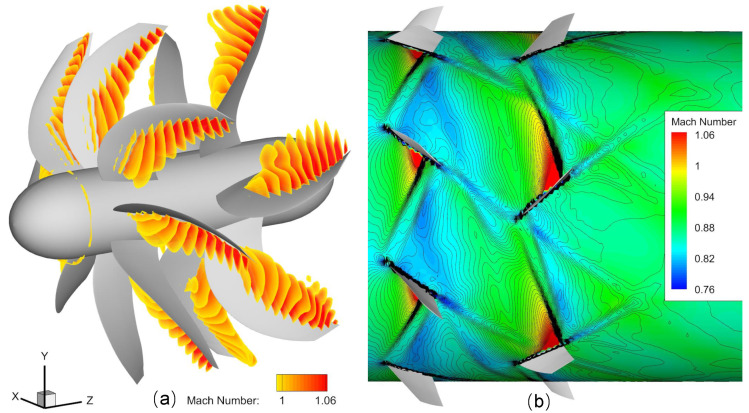
Simulation results of shock waves in open rotor blade passage. (**a**) Range of supersonic flow, (**b**) Mach number distribution on the section at 86% blade height.

**Table 1 entropy-25-00432-t001:** Coefficient of drag $\bar{C_{d}}$ and non-dimensional wake bubble length $L_{w}/D$ for steady flow past a cylinder at Re = 40.

Cases	$\bar{C_{d}}$	$L_{w}/D$
Jost and Glockner [35]	1.59	2.25
Russel and Wang [36]	1.60	2.25
Xu and Wang [37]	1.66	2.21
Tseng and Ferziger [3]	1.53	2.21
present study	1.58	2.24

**Table 2 entropy-25-00432-t002:** Coefficient of $\bar{C_{d}}$ and St for steady flow past a cylinder at Re = 40 with different $\Delta x_{min}$.

$\Delta x_{\min}$	$\bar{C_{d}}$	St
0.08D	1.268	0.190
0.04D	1.277	0.192
0.02D	1.289	0.195
0.01D	1.290	0.195

**Table 3 entropy-25-00432-t003:** Numerical and experimental values of ${\bar{C}}_{d}$, $C_{Lr.m.s.}$ and St at Re = 185 (non-oscillating cylinder).

Cases	${\bar{C}}_{d}$	$C_{Lr.m.s.}$	St
Experimental results [39]	1.280	—	0.190
Lu and Dalton [40]	1.310	0.422	0.195
Guilmineau and Queutey [39]	1.287	0.443	0.195
Khalili et al. [2]	1.282	0.431	0.191
Liu and Hu [41]	1.289	0.451	0.197
Wu [42]	1.267	0.468	0.195
Present results	1.290	0.445	0.195

**Table 4 entropy-25-00432-t004:** Comparison of computed data for flow at Re =185, $f_{e}/f_{0}$ = 0.8.

Cases	${\bar{C}}_{d}$	$C_{dr.m.s.}$	$C_{Lr.m.s.}$
Uhlmann [45]	1.354	—	0.166
Guilmineau and Queutey [39]	1.195	0.036	0.08
Yang et al. [46]	1.281	0.042	0.076
Schneiders et al. [44]	1.279	0.042	0.086
Khalili et al. [2]	1.287	0.045	0.079
Present results	1.190	0.043	0.084

**Table 5 entropy-25-00432-t005:** Comparison of computed data for flow at Re =185, $f_{e}/f_{0}$ = 1.1.

Cases	${\bar{C}}_{d}$	$C_{dr.m.s.}$	$C_{Lr.m.s.}$
Guilmineau and Queutey [39]	1.420	0.149	0.897
Wu [42]	1.418	—	0.906
Present results	1.408	0.141	0.910

**Table 6 entropy-25-00432-t006:** Nodes numbers of Rotor mesh and Cartesian mesh.

Mesh Blocks	Node Number	Block Number	Total Node Number
Front rotor	0.204 million	8	1.632 million
Rear rotor	0.204 million	6	1.224 million
Cartesian mesh			4.194 million

**Table 7 entropy-25-00432-t007:** CPU time consuming ratio of OGA.

Cases	Cylinder	Open Rotors
Cartesian Mesh Solving	46%	11%
Body-fitted Mesh Solving	51%	75%
Overset grid assembly	3%	14%

**Table 8 entropy-25-00432-t008:** Comparison of computation time between two OGA methods.

OGA Method	Cylinder Case	Open Rotors Case
traversal method	43.56 s	215,325 s
local inverse mapping method	0.24 s	25 s

## Data Availability

Not applicable.

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
