# Peer review of "Local Inverse Mapping Implicit Hole-Cutting Method for Structured Cartesian Overset Grid Assembly"

_entropy, 2023, doi:10.3390/e25030432_

Round 1

Reviewer 1 Report

As a mathematician, I accepted to review the paper because seemed it is about solving partial differential equations. so perhaps I am not a right person to evaluate this manuscript.

After reading the paper it looks like a report on some results of some code for which it is not clear in what sense it is novel. The result seem to be consistent with previous, thus expected. Unfortunately, I do not see what is the contribution of this work. I would expect either clear and detailed outline of the novelty of the algorithm(s) (or coding) or/and some data how much faster the new algorithm is (compared to the existing, available).

Author Response

Point 1: I would expect either clear and detailed outline of the novelty of the algorithm(s) (or coding) or/and some data how much faster the new algorithm is (compared to the existing, available).

Response 1:  In Section 4.3, the comparison result of calculation time between LIM method and traversal method is added. The description of flux interpolation and parallel computation strategy of LIM method is added.

OGA computing is implemented on a personal computer with AMD Ryzen 3100 CPU. The traversal method and LIM method were used respectively to mark the FPNS Cartesian mesh points. The comparison of OGA calculation time is shown in Table 8. The comparison results show that the LIM method can reduce the computing time by 99% compared with the traversal method. Compared with 2-D case, the computational efficiency of LIM method is significantly improved in 3-D case. 

Reviewer 2 Report

This article has a clear logical structure and proposes a novel approach which can solve the problem more efficiently,local inverse mapping implicit hole-cutting method (LIM),to the idea.Some sentences in the article have grammatical errors that should be corrected.

Author Response

Point 1: Some sentences in the article have grammatical errors that should be corrected.

Response 1: Grammar and spelling errors were carefully checked and corrected.

Reviewer 3 Report

In this paper, a local map method is proposed for overlapping Cartesian grid and body-fitted grid, and high-order scheme is used to simulate the flow field on Cartesian grid.

1) The topic of this paper is the overlapping grid assembly (OGA) technology, however the introduction of the OGA itself is insufficient.

2) In the inverse map method, a homogeneous Cartesian grid is used to improve the donor cell searching efficiency. This is a relatively mature method and is lack of innovation.

3) The complexity of overlapping grid method is reflected in multi-body problem, high-order conservation interpolation, and parallel assembly. Can the method in this paper be applied to the multi-body problem? The example shown is too simple.

4) Grid information, test environment, time of solver and overlapping assembling is lacked in the test cases. Is parallel computing considered?

Author Response

Point 1:  The topic of this paper is the overlapping grid assembly (OGA) technology, however the introduction of the OGA itself is insufficient.

Response 1:  The research progress of OGA method is added in the introduction part. The Wall-Distance method reduces the number of donor cells to be searched, thus reducing the amount of computation. When the number of Grid nodes exceeds 10 million, parallel computing method should be adopted for Overset Grid Assembly to improve computing efficiency. 

Point 2:  In the inverse map method, a homogeneous Cartesian grid is used to improve the donor cell searching efficiency. This is a relatively mature method and is lack of innovation.

Response 2: Although the LIM overset grid assembly method proposed in this paper is relatively simple, it can effectively improve the search efficiency of the donor cells in the three-dimensional body-fitted mesh block. The calculation time comparison with traversal method is added in section 4.3.1, and the results are shown in Table 8.

Point 3:  The complexity of overlapping grid method is reflected in multi-body problem, high-order conservation interpolation, and parallel assembly. Can the method in this paper be applied to the multi-body problem? The example shown is too simple.

Response 3: LIM method can be applied to the multi-body problem. Take the open rotors case as an example, the front and rear rotors rotate in opposite directions. Each blade grid block can move independently, and there are 14 moving blocks in total. This example demonstrates the applicability of the LIM method to the multibody case. In section 4.3.2 and 4.3.3, the flux interpolation strategy and parallel computing strategy are introduced.

Point 4:  Grid information, test environment, time of solver and overlapping assembling is lacked in the test cases. Is parallel computing considered?

Response 4:  In the newly added section 4.3.1, the number of grid nodes and the computing environment are introduced, and the calculation time of OGA solution is given.

Round 2

Reviewer 1 Report

OK, I have no objection, act according to decisions of fellow editors.